# Characterization of Monoamine Oxidase-A in tropical liver fluke, *Fasciola gigantica*

**Mirza Ahmar Beg**[ID]*****, Abdur Rehman, Lubna Rehman, Rizwan Ullah, Faiza Farhat, Sobia Wasim, S. M. A. Abidi**

Section of Parasitology, Department of Zoology, Aligarh Muslim University, Aligarh, India

\* ahmarbeg@gmail.com

**Data Availability Statement:** All relevant data are within the paper and its Supporting information files.

**Funding:** The authors received no specific funding for this work.

## Abstract

*Fasciola gigantica*, responsible for the zoonotic disease fasciolosis, pose a great threat to the livestock and human health worldwide. The triclabendazole (TCBZ) has been used for decades as a broad spectrum anthelmintic to control this perilous disease but the emergence of resistance in flukes against TCBZ has prompted researchers across the world to explore for new drugs and antigenic targets. World Health Organization has strongly recommended the utilization of neurobiologically significant biomolecules as new drug/antigenic targets because of their significant role in the physiology of parasites. Monoamine Oxidase (MAO) is an important neurobiological enzyme which catabolizes aminergic neurotransmitters thus preventing prolonged excitation of neurons and in non-neuronal cells it prevents cellular toxicity due to accumulation of toxic monoamines. Owing to the important role of MAO in the survival and perpetuation of parasites, multipronged approaches were undertaken for the characterization of MAO-A in *F. gigantica*. The activity of MAO was found to be 1.5 times higher in the mitochondrial samples than the whole homogenate samples. The adult worms of the *F. gigantica* appeared to possess both the isoforms of MAO i.e., MAO-A and MAO-B. The zymographic studies revealed strong enzyme activity in its native state as assessed through prominent dark bands at 250KDa in the zymogram. The enzyme was also found to be highly immunogenic as revealed by high antibody titer at 1:6400 dilution. The immunogenicity of MAO-A enzyme was further established in the Western Blots in which a strong band of 50KDa was distinctly evident. Despite ubiquitous presence of MAO in *F. gigantica* some regions like tegumental surface and intestinal caecae displayed strong immunofluorescence as compared to other regions. The detection of MAO-A in the *F. gigantica* samples in Dot-Blot assay indicate a great potential of this molecule for the immunodiagnostics of fasciolosis, particularly in the field conditions. The enzyme activity was sensitive to the specific inhibitor clorgyline in a concentration dependant manner, particularly in the late incubation period. The zymographic results also exhibited similar trend. The strong intensity of spots in Dot-blots indicate high immunogenicity of the MAO protein. The intensity of bands/spots in the samples of worms treated with clorgyline also declined, clearly indicating that the tropical liver fluke possesses prominent MAO-A activity.

**Competing interests:** The authors have declared that no competing interests exist.

## Introduction

Fasciolosis, caused by a digenetic trematode *Fasciola gigantica*, is a menacing disease which invokes huge monetary loss to agrarian based economies which is estimated to be more than 3 billion USD annually [1]. Besides fiscal losses it causes serious pathological conditions to their hosts and even fatality at high parasitic burden. *Fasciola* infection is highly prevalent in Indian subcontinent and across the world conspicuously in tropical and subtropical countries of Asia, Africa and Europe [2]. A prevalence of 31.14% in the North Indian region correlates with the reason behind high morbidity amongst the cattle in these areas [3, 4]. Though fasciolosis primarily affects cattle, its zoonotic potential cannot be neglected [5, 6]. From India alone there have been numerous reports of human *Fasciola* infection from different states such as Arunachal Pradesh, West Bengal, Assam, Uttar Pradesh etc. [7–12]. Hence, there is a growing need to address the problems related to the *Fasciola* infection in livestock and other probable hosts including humans.

In the past few decades notable work has been done to identify antigens of various classes as vaccine candidates and efforts have been made to formulate drugs against these specific targets. However the overuse of most widely used drugs including Triclabendazole has compromised their efficacy [13] and eventually a growing drug resistance has been observed in these flukes [14, 15] which has prompted researchers across the globe to identify and utilize various antigens as immunogenic and vaccine targets such as cysteine proteases, antioxidant system, tegumental biomolecules, Fatty acid binding proteins [16–22] etc. but neurobiological system remained neglected as antigenic targets. Owing to their significant roles in parasite biology neurobiological targets could be effectual candidates for vaccine development and drug designing as a number of neuropeptides and amine metabolizing enzymes are directly involved in motility, control and coordination of the worms [23]. The key problems like migration of worms, attachment to host and obtaining nutrition are highly reliant on the neuromuscular system. Therefore, it was suggested by World Health Organization that there is a dire need to study the antigenic targets for anthelmintic drugs associated with neurophysiological aspects [24].

Due to the important connecting position in evolutionary lineage in relation with central nervous system, flatworms particularly *Fasciola hepatica* were the first organisms to be studied about the mechanisms of serotonin in relation to the rhythmic movements, glucose uptake, glycogenolysis and lactate production of flukes, thus opening the new avenues related to neurobiology [25–27]. Serotonin, one of the prime MAO substrates is of great significance in the helminths as the oral sucker is deeply enriched with serotonergic neurons which plays a pivotal role in the attachment of worms with the host [28, 29]. The migratory behavior of helminth is also highly dependent on the circadian levels of serotonin as shown in adult *H. diminuta* [30]. Researchers have also demonstrated about the significant role of serotonin in the mating behavior of *Ascaris* [31] which shows that MAO and its substrate plays a key role in host-parasite interface and perpetuation of life cycle of helminths.

Parasitic helminths can be broadly categorized into phylum Platyhelminthes which comprises of flatworms and phylum Nemathelminths, which primarily encompasses roundworms. Individuals of these phyla lacks a well-defined circulatory system, so it becomes a challenging task for them to maintain the bodily balance of peptide and aminergic neurotransmitters and hormones which are involved in the control and coordination of these parasites [23]. For this purpose, Monoamine Oxidase (MAO) plays a crucial role. In neurons it is gathered in large quantity at the inter-neuronal junctions acting as monoamines scavenger thus preventing prolonged excitation of neuronal cells. In non-neuronal cells, it prevents accumulation of monoamines which proves to be toxic for cell if their concentration grows past the threshold level.

These monoamines are formed by side reaction of main synthetic pathway or exogenous amines derived from dietary sources [32].

Monoamine oxidase is a flavin containing enzyme found in the outer membrane of mitochondria. It belongs to oxidoreductase class of enzymes which causes oxidative deamination of dietary, monoaminergic neurotransmitters and hormones [33, 34]. On the basis of their substrate and inhibitor sensitivity, two different forms of MAO have been identified [35]. Evidence from immunological studies using monoclonal antibodies and inhibitor sensitivity supports the existence of two different forms of the enzyme and has permitted separation of MAO-A and MAO-B in organisms as advanced as humans [36] and as primitive as helminths [23]. While MAO-B is involved in general coordination of worms, MAO-A is primarily responsible for aggression and motility of the parasites thus making it an important molecule for drug target and vaccine development.

The abundance of MAO in the helminthic system could be attributed to the amine rich diet of helminths as obtained from their hosts which needs to be neutralized on a regular basis [37]. MAO also plays a cardinal role in the development of helminths as also seen in case of *A. galli* [38]. The levels of MAO and its associated substrate have been found to greatly affect important life functions such as growth promotion [39], reproduction and egg production [40]. Until now, MAO has been reported in parasitic helminths belonging to different classes, namely, *Schistosoma mansoni* [41], *Hymenolepis diminuta* [42], *Ascaris lumbricoides* [43] and some other helminth species [44–49] but an in-depth study involving multifrontal approaches in a particular parasite is still a void which is needed to be filled.

Therefore, the present study is an attempt to characterize MAO-A in adult *F. gigantica* using multipronged approaches involving biochemical and immunological aspects. Further, effect of Clorgyline, a MAO-A inhibitor was studied on the *F. gigantica* worms under *in vitro* conditions in order to find relation between the motility of worms with that of MAO-A. Previously researchers have reported MAO in helminths but a multi-omics study involving holistic approach in a particular parasite was lacking.

## Materials and methods

### Parasite collection

Adult *F. gigantica* worms were procured from the infected liver of buffaloes collected from the local abattoirs of Aligarh region, India. Worms were rinsed in Hanks' Balanced Salt solution (HBSS), pH 7.4, pre-maintained at 37˚C to remove host contaminants. The worms were then rinsed with antibiotic & antimycotic solution to ward off bacterial and fungal contaminants. Parasites were again given a quick rinse with HBSS, damp dried on Whatman® filter paper, used either fresh or stored at -80˚C in deep freezer (Haier) for further studies.

### Identification of *Fasciola gigantica* using ITS2 sequences

**DNA extraction.** DNA of parasites was extracted by the QIAamp DNA Mini kit (QIAGEN GmbH, Hilden, Germany) as per the standard protocol as described by the manufacturer. A total of 0.1g of each parasite was taken for the extraction of DNA which yielded enough DNA for carrying out the PCR using the specific primers. The DNA concentration at 260 nm and purity at 260:280 nm was checked using Biophotometer (Eppendorf, USA).

**Amplification of DNA using polymerase chain reaction.** The PCR was performed in 25μl reaction mixture containing 1μl template DNA (500ng), 12.5μl of PCR master mix, 2.5μl forward and reverse primers (1μM) and 6.5μl nuclease free water to give a total volume of 25μl in PCR tubes. The primers were designed to target second internal transcribed spacer as per the specific primers as mentioned below:

*Fasciola gigantica* (forward primer): 5′–**GTGCATTTTAGCAACTCGCA**–3′

*Fasciola gigantica* (reverse primer): 5′–**AAACGCCATAGATCTGGCAC**–3′

PCR was performed on a gradient thermal cycler (BioRad) under the following thermal cycling conditions for DNA template of *F. gigantica*:

Initial denaturation at 95°C for 15 min; then 35 cycles of 95°C for 1min, 43.1°C for 1 min and 72°C for 1 min; 72°C for 10min.

The amplified PCR products were electrophoresed on a 1.5 per cent agarose gel prepared in Tris EDTA (TE) buffer containing ethidium bromide and visualized on a UV transilluminator (Genetix, India).

## Homogenization of *F. gigantica* samples

A homogenate of 25% consistency (w/v) was prepared to obtain high yield of mitochondria. The freshly obtained adult flukes were carefully weighed and minced in 50mM Tris-HCl buffer containing 0.25M sucrose, 1.0mM ATP, 1.0mM EDTA, 1.0mM $MgCl_2$, 100mM KCl and 0.5% bovine serum albumin (BSA) using the tissue homogenizer with glass-teflon probe fitted with a motor driven pestle. All the procedural activities were carried out at 4°C.

## Mitochondrial isolation

Mitochondria were isolated by differential centrifugation method as described by Podesta *et al*. [50]. Primarily, the parasites were homogenized in phosphate buffer and centrifuged two times at 1000g for 15 minutes to remove cell debris. The supernatant was collected and centrifuged at 8000g for 30 minutes. Mitochondrial pellet thus obtained was resuspended in homogenization medium with Bovine Serum Albumin and centrifuged at 8000g for 30 minutes. The pellet was again resuspended in medium without BSA followed by centrifugation at 16000g for 30 minutes. Washing of pellet was done with tris-HCl, pH 7.4 by resuspending and centrifuging the pellet at 16000g for 30 minutes. The pellet obtained was stored at -80°C for further studies. All the steps of centrifugation were done at 4°C to keep the activity of enzyme intact.

## Protein estimation

The protein content in the whole homogenate as well as mitochondrial fractions of the adult *F. gigantica* worms was estimated using the dye binding method as described by Spector [51] using BSA as standard.

## Quantitation of MAO activity in *F. gigantica*

Activity of Monoamine Oxidase was quantified using the method described by Tabor *et al*. [52]. A final volume of 2 ml assay mixture consisted of 100mM phosphate buffer (pH 7.2), 5.0 mM tyramine hydrochloride, double distilled water and isolated mitochondrial pellet. Assay mixture was incubated at 37°C, after 30 minutes of incubation reaction was stopped by adding 1ml of 10% perchloric acid. The assay mixture was read at 250nm in spectrophotometer (Taurus Scientific). The control assay mixture was devoid of substrate. Specific enzyme activity was expressed as μmoles tyramine utilized/mg protein/30minutes at 37°C.

## Quantification of MAO-A and MAO-B activity in *F. gigantica*

The activity of MAO-A and MAO-B, the two isoforms of MAO, was analyzed as described earlier following the method of Tabor *et al*. [52]. In this study, the mitochondrial isolates were

separately treated with 60µM concentration each of Deprenyl and Clorgyline for 30 minutes at 37˚C and the activity was measured using Benzylamine, the substrate catabolized by both the isoforms of MAO. The control was devoid of any substrate. Enzyme activity was expressed as µmoles benzylamine utilized/mg protein/30 minutes at 37˚C.

## Zymography

For the visual evidence of enzyme activity, zymography was done as described by Lee *et al*. [53] with slight modifications. The Homogenate of samples were first incubated in 0.5% Triton-X100 for 120 minutes at 4˚C. A total of 10µg protein were loaded in each well with an equal volume of sample buffer without β-mercaptoethanol and were resolved in 8% polyacrylamide gel without SDS at a constant voltage of 90V for 90 minutes at 4˚C. The gel was then incubated in a substrate prepared by dissolving Tyramine-HCl (20mg), Nitroblue tetrazolium (10mg), 0.1MTris-HCl (15ml) and distilled water (10ml) for 180 minutes in an incubator maintained at 37˚C.

## Enzyme Linked Immunosorbent Assay (ELISA)

To determine the immunogenic potential of MAO, the ELISA was performed by the method of Voller *et al*. [54]. Round bottomed polystyrene microtiter plates were coated with 50µl homogenate (10µg/ml protein) of *F. gigantica* and buffalo liver except blank which was coated with 50µl PBS only. After that washing was done with Tris-buffered saline with Tween (TBST) 5 times for 5 minutes each. Blocking of unspecific sites was done with blocking buffer for 120 minutes. The anti-MAO-A primary antibody (Sigma Aldrich) was serially diluted with an initial dilution of 1:50 and allowed to react with antigens overnight at 4˚C for proper adherence. Washing was done again with TBST 5 times for 5 minutes each to wash away any unbound primary antibody. Thereafter, alkaline phosphatase conjugated anti-rabbit IgG (secondary antibody) was added to each well in a dilution of 1:10000 for 120 minutes followed by washing with TBST. Substrate solution was prepared by dissolving pNPP and Tris-HCl tablets (SIG-MAFAST™) in distilled water. The reaction gave dull yellow color upon addition of substrate and the O.D was recorded on ELISA plate reader at 415 nm wavelength.

## Sodium Dodecyl Sulphate-Polyacrylamide Gel Electrophoresis (SDS-PAGE)

The proteins were resolved in a 12% Polyacrylamide gel containing SDS following standard method of Laemmli [55]. To concise, 25mg protein was loaded in each well and electrophoresis was done at a constant voltage of 90V till the tracking dye has reached the bottom of gel. Thereafter the gel was stained with Coomassie Brilliant Blue Dye CBBR-250 and imaging was done.

## Western blotting

To determine the molecular weight of the immunogenic polypeptides, the western blotting was performed. First, the proteins were resolved on a 12% SDS-Polyacrylamide Gel and transferred to methanol activated PVDF membrane by the method of Towbin *et al*. [56] as described by Khan *et al*. [20]. Briefly, after the transblotting of protein to the PVDF membrane, the membrane was kept in blocking buffer for 120 minutes for blocking the unoccupied sites. Thereafter, the membrane was incubated overnight in rabbit anti-MAO-A antibody (primary antibody) at 4˚C with a dilution of 1:1000 in blocking buffer. The membrane was then washed with TBST and incubated with alkaline phosphatase conjugated anti-rabbit secondary

antibody for 120 minutes at room temperature with a dilution of 1:10000. The membrane was again thoroughly washed with TBST. Finally, membrane was incubated in substrate solution prepared by dissolving BCIP/NBT tablets (SIGMAFAST™) in 20ml distilled water until the color developed. Termination of experiment was done by washing the membrane with distilled water multiple times. Images of air-dried membrane were captured and analyzed.

## Immunohistochemistry

IHC was done according to the method as described by Khan *et al*. [20]. The worms were flat fixed in buffered formalin for 24 hours followed by dehydration in the ascending grades of ethanol. The worms were then cleared in xylene and finally kept in molten paraffin wax at 72 ± 1ºC for 12 hours in a water bath. The blocks of wax containing the worm in the middle were prepared and about 5µm thick sections of parasite were cut using rotary microtome (Yorco, India). The sections were deparaffinized in xylene and rehydrated in descending grades of ethanol. Antigen retrieval was done afterwards by boiling the sections in citrate buffer in a microwave for 15 minutes at full power. After that blocking of sections was done using the blocking buffer and then incubated overnight at 4˚C in anti-MAO-A antibodies (1˚Ab) used at a dilution of 1:1000 in the blocking buffer. Sections were then washed again with TBST 5 times and incubated in FITC conjugated anti-rabbit secondary antibody used at a dilution of 1:10000 for 120 minutes at room temperature. Imaging of sections was done on a Confocal microscope (Zeiss LSM-780) maintained at University Sophisticated Instrumentation Facility, Aligarh Muslim University, India.

## Dot blot assay

This immunodiagnostic technique was performed following the method of Hawkes *et al*. [57]. A 10% Homogenate (w/v) was prepared using parasite samples from bovine, ovine and piscine origin: *Fasciola gigantica*, *Gigantocotyl explanatum*, *Gastrothylax crumenifer*, *Paramphistomum epiclitum* and *Clinostomum complanatum* respectively. The homogenates were incubated with 0.5% Triton-X100 for 2 hours at 4˚C. A fixed quantity of homogenate (5µl) from each sample was put on preactivated PVDF membrane and left to dry. After that membrane was blocked using blocking buffer for 120 minutes followed by overnight incubation in rabbit anti-MAO-A antibody (1˚ Ab) with a dilution of 1:1000 at 4˚C. After 12-hour incubation, primary antibody was discarded and membrane was washed 5 times with TBST for 5 minutes. Thereafter, membrane was treated with alkaline phosphatase conjugated IgG (secondary antibody) for 120 minutes at room temperature and again washed with TBST 5 times. Membrane was then incubated in substrate solution prepared by dissolving BCIP/NBT tablets (SIGMAFAST™) in 20ml distilled water until the color developed and reaction was stopped by washing membrane with distilled water several times.

## Treatment of parasites with inhibitor

A total of 5 worms were incubated in 25ml solution containing clorgyline,a specific inhibitor of MAO-A in three different concentrations of 30µM, 60µM and 90µM. Control group was devoid of any inhibitor. The worms were incubated for 7 hours in a $CO_2$ incubator maintained at 37˚C. All the treatments were given in triplicates.

**Worm motility.**   After the incubation, the motility of the parasites was recorded as per the method of Stepek *et al*. [58] with an interval of 30 minutes with a scale of 0–7 where 0 denotes complete loss of motility; 1 signifies movement in worms only when prodded; 2 indicates worms active only at extreme ends of the body; 3 means overall sluggish movement of the entire body; 4 depicts active worms throughout and 5 shows highly motile worms. Above 5

marks extreme motility with intense wriggling movements. The MAO-A activity of clorgyline treated worms was assessed as described earlier according to the method of Tabor *et al.* [52]. To provide visual evidence of change in enzyme activity post treatment of worms with varying doses of clorgyline, zymography was performed according to the method of Lee *et al.* [53]. To check the alteration in the immunogenicity of clorgyline treated worms, Western Blotting and Dot-Blot assay was also performed according to the method of Towbin *et al.* [56] and Hawkes *et al.* [57] respectively, as described earlier.

**Statistical analysis.** Statistical analysis was done using One way ANOVA where a minimum of three replicates were performed for each experiment. The results are showed as Mean ± SEM ($p \leq 0.05$ considered as significant).

## Results

The results of the present study are summarized in Figs 1–7 and Table 1.

The model digenetic trematode parasite, *F. gigantica*, selected for the present study was correctly identified on the basis of both, morphological as well as molecular characters and the enzyme understudy was found to be highly enriched in the mitochondrial fraction obtained from the total homogenate of the adult worms. The parasite selected for the present investigations was the tropical liver fluke, *Fasciola gigantica* since a single specific amplicon band was obtained using *F. gigantica* specific primers following the PCR (Fig 1a).

### Characterization of MAO in adult tropical liver fluke, *Fasciola gigantica*

The specific activity of MAO was determined using tyramine as the substrate and it was found to be about 1.5-fold higher in the mitochondrial fraction as compared to the activity in the whole homogenate of the adult worms (Fig 2a). The use of MAO-A and MAO-B specific inhibitors, Clorgyline and Deprenyl respectively, showed that there was a significant decrease in the overall mitochondrial MAO activity of *F. gigantica* as compared to the control, which further confirmed the existence of two isoforms of MAO in *F. gigantica* (Fig 2b). Further, zymographic analysis revealed the presence of a single distinct band of about 250KDa (Fig 3a

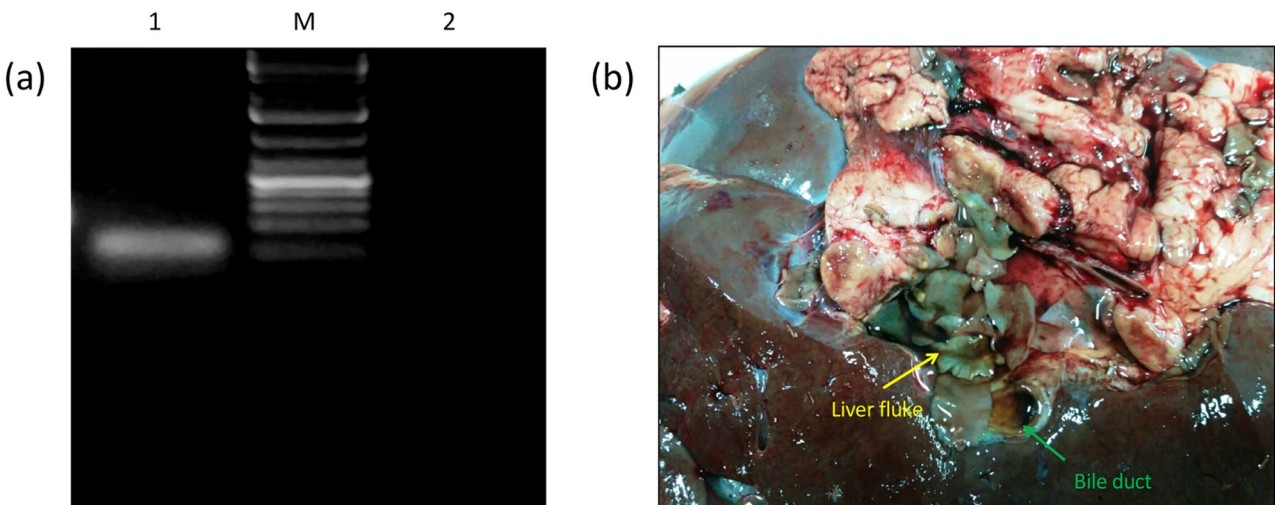

**Fig 1.** Representative image (a) of amplified DNA obtained following PCR by using primers of *F. gigantica* ITS2 region (lane 1), non template control (lane 2) and DNA ladder of 1kb size (M). A distinctive band of amplicon could be observed in lane 1, suggesting the identification of sample as the tropical liver fluke (b).

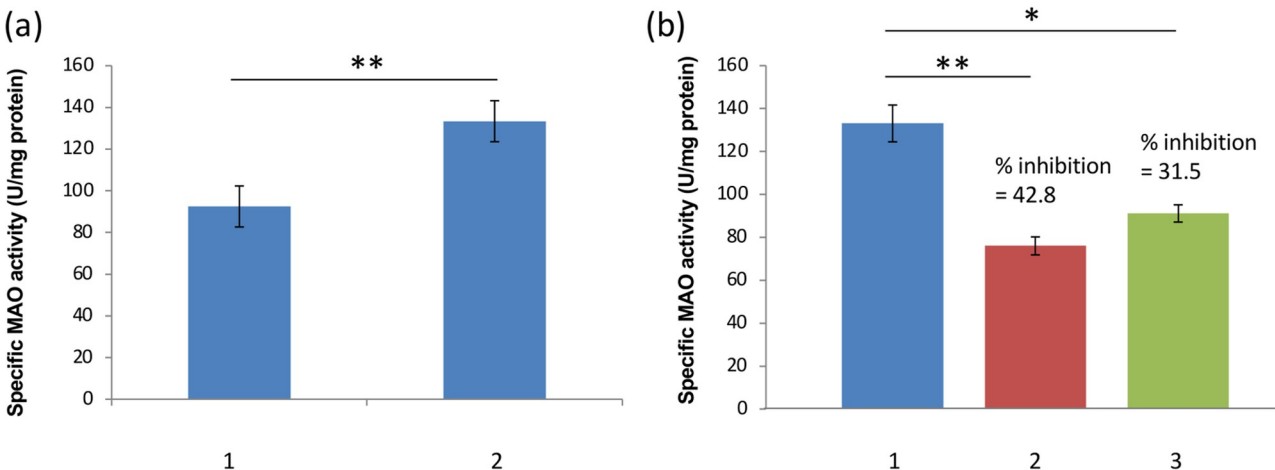

**Fig 2.** (a) The level of Monoamine Oxidase (MAO) activity of *F. gigantica* in the whole homogenate (1) and the mitochondrial isolate (2). Since MAOs are present on the outer mitochondrial membrane the enzyme activity was found to be significantly higher (about 1.5-fold increase) in the mitochondrial fraction as compared to the homogenate. The specific enzyme activity was expressed as μmoles Tyramine utilized/mg protein/30minutes at 37˚C. (b) The specific enzyme activity of Monoamine Oxidase (MAO) in the mitochondrial isolate of *F. gigantica*. 1: *F. gigantica* control group devoid of any inhibitor. 2: *F. gigantica* mitochondrial isolate treated with clorgyline. 3: *F. gigantica* mitochondrial isolate treated with deprenyl. There was a significant decrease in enzyme activity as compared to control when selective inhibitors of MAO were used separately. A minimum of 3 replicates were run and the data is expressed as a mean ± SEM.

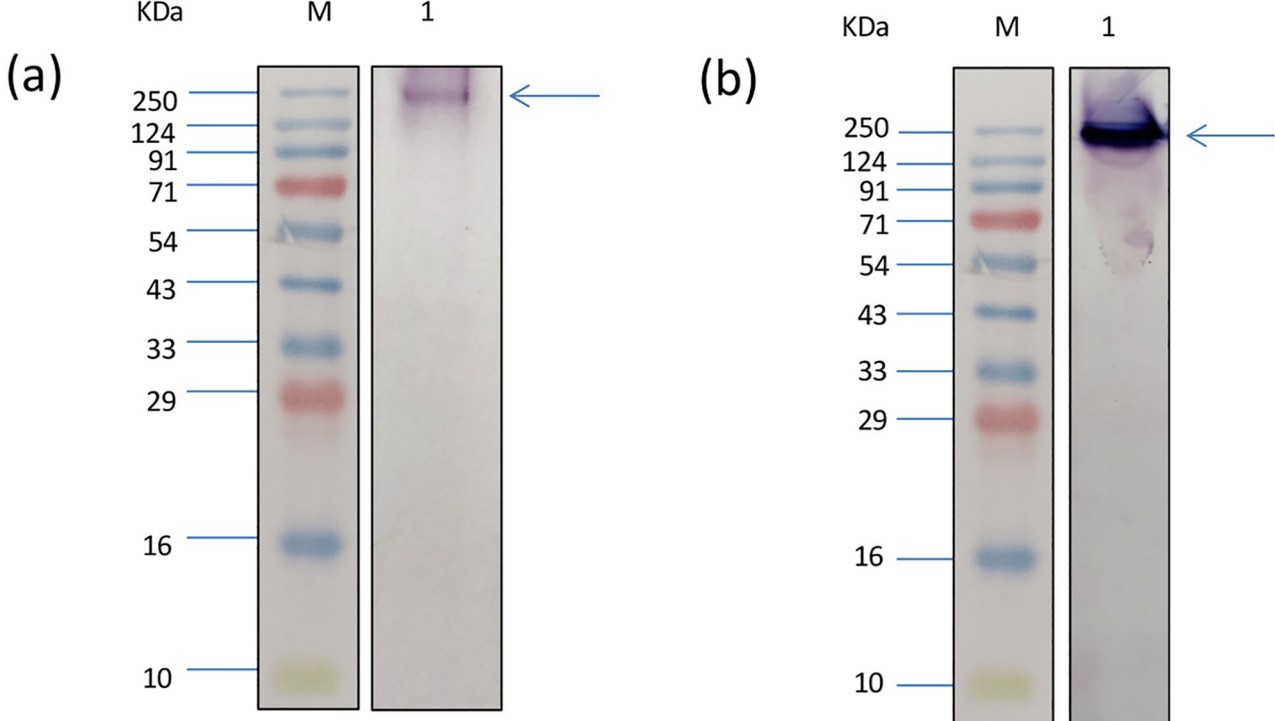

**Fig 3.** A single band of about 250 kDa was obtained when the whole homogenate of *F. gigantica* (a) and mitochondrial fraction (b) were subjected to zymography using Tyramine-HCl and Nitroblue tetrazolium. M: standard molecular weight marker.

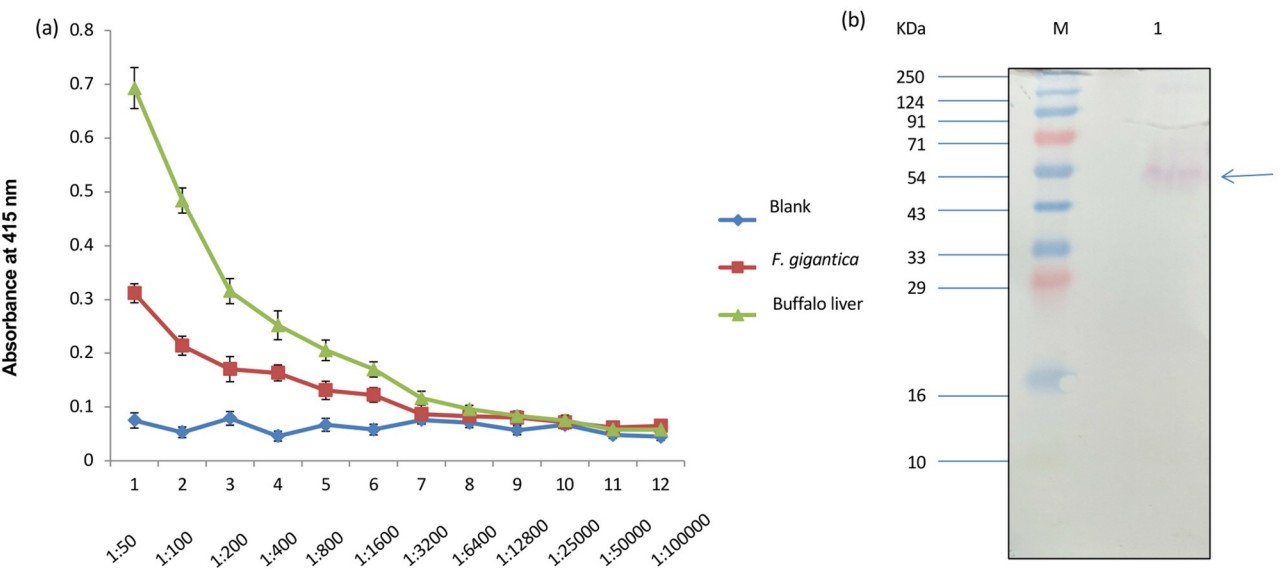

**Fig 4.** MAO-A was found to be highly immunogenic as revealed by high antibody titre following indirect ELISA (a) using monospecific anti MAO-A polyclonal antibodies. A distinct 50 kDa polypeptide band was also detected in the western blots (b) performed using whole homogenate (lane 1). M: standard molecular weight marker.

and 3b). For the immunological characterization of MAO-A in *F. gigantica* worms a number of studies were performed. The indirect ELISA test performed using the mono specific anti-MAO-A polyclonal antibodies showed strong immunogenic nature of the MAO-A enzyme of the adult *F. gigantica* worms as evident from high antibody titer at 1:6400. The buffalo liver sample was used as a positive control which exhibited significantly higher values (Fig 4a) and the titer appeared to be much higher, almost at 1:12800.

A single polypeptide band obtained of about 50KDa obtained in the SDS polypeptide profile was also detected on the western blots using anti-MAO-A antibodies, thus confirming the presence of MAO-A in the parasite samples beside reflecting the immunogenic potential of the mitochondrial MAO-A of *F. gigantica*. A single band on the Western blots also reflects that the isoforms might be existing in the monomeric form (Fig 4b). However, further studies are required to ascertain the number and size of different isoforms through molecular characterization.

The immunolocalization of MAO-A in the tissue sections of adult *F. gigantica* worms using anti-MAO-A antibody. The confocal micrographs revealed ubiquitous localization of MAO-A in the musculature, tegument, hypodermis, intestinal caecae and vitellaria but the immunofluorescence was maximum around intestinal caecae, hypodermis and tegument, the regions through which the parasite obtains nutrition and areas of absorption as well as of high metabolic activity which may help the parasite to survive in their micro-habitat and also help in maintaining the host-parasite relationship (Fig 5a).

The results of dot blot assay revealed that the samples of different parasites of bovine and ovine origin including *F. gigantica* were found positive for the presence of MAO-A which needs to be worked out in future for better understanding of role of MAO in relation with the habitats of different parasites. Most intensely colored dot appeared in case of positive control (buffalo liver) followed by *F. gigantica* and *P. epiclitum* while *C. complanatum* showed very low signal as compared to liver (Fig 5b and 5c). The negative control was devoid of any antigen (4) but conclusive statements cannot be made for the presence of MAO in *C. complanatum*

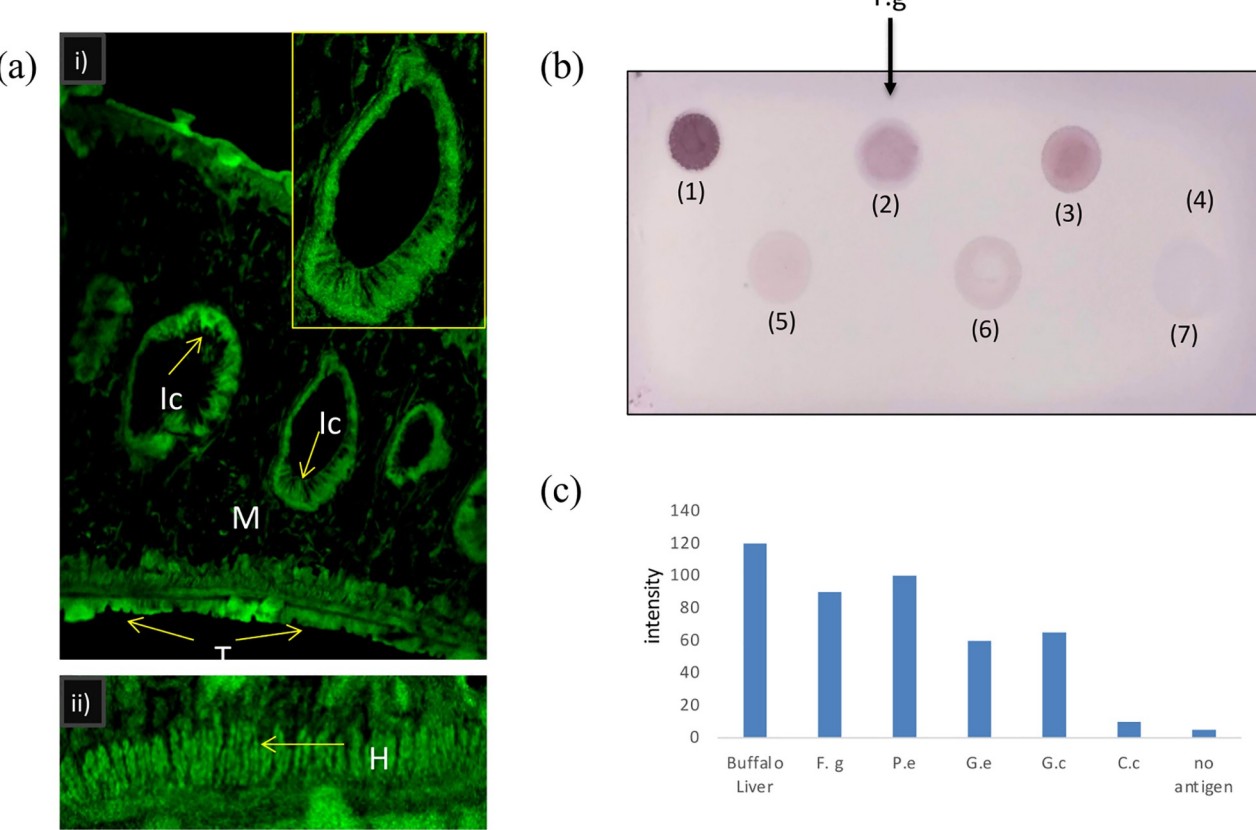

**Fig 5.** Confocal micrographs of tissue sections of adult tropical liver fluke *F. gigantica* (a). MAO-A was predominetly localized in the intestinal caecae, tegument, musculature and hypodermis using anti MAO-A polyclonal antibodies (i), inset showing magnified image of intestinal caecae. A magnified micrograph showing immunolocalization in hypodermis region (ii). Scale~200µm. (T- tegument, Ic- Intestinal caecae, H- hypodermis, M-musculature). (b) Representative dot blot with most intensely colored dot appeared in case of buffalo liver (1) followed by *F. gigantica (2)* and *P. epiclitum (3)* while *C. complanatum (7)* showed very low signal as compared to liver. The negative control was devoid of any antigen (4). (c) Quantification of intensity of samples of Dot-blot assay.

unless verified through other biochemical studies because it is possible that the antibody used in the present study might not have cross-reacted with sample from fish parasite, also a reflection of the species-specific variation in enzyme molecules with respect to the cross-reactive epitopes.

## Effect of clorgyline treatment on adult *F. gigantica* worms

The motility of parasites incubated in different concentrations of clorgyline followed a definite trend where the motility of worms first increased initially up to 3–4 hours in a dose dependant manner and then declined subsequently followed by complete immobility leading to paralysis of worms (Fig 6a). It was interesting to observe that contrary to the direct effect of clorgyline on the mitochondrial MAO-A of *F. gigantica*, the intact worms exhibited stimulatory response in motility proportional to the concentration of inhibitor during the early phase of incubation of worms which was followed by complete loss of motility in late incubation hours.

The specific activity of the mitochondrial MAO after incubation of worms with clorgyline was inhibited in a dose dependant manner. As compared to control, the 60µM and 90µM doses of clorgyline significantly inhibited the MAO-A activity but changes were not significant at the 30µM dose (Fig 6b and Table 1). Moreover, zymographic analysis of the mitochondrial

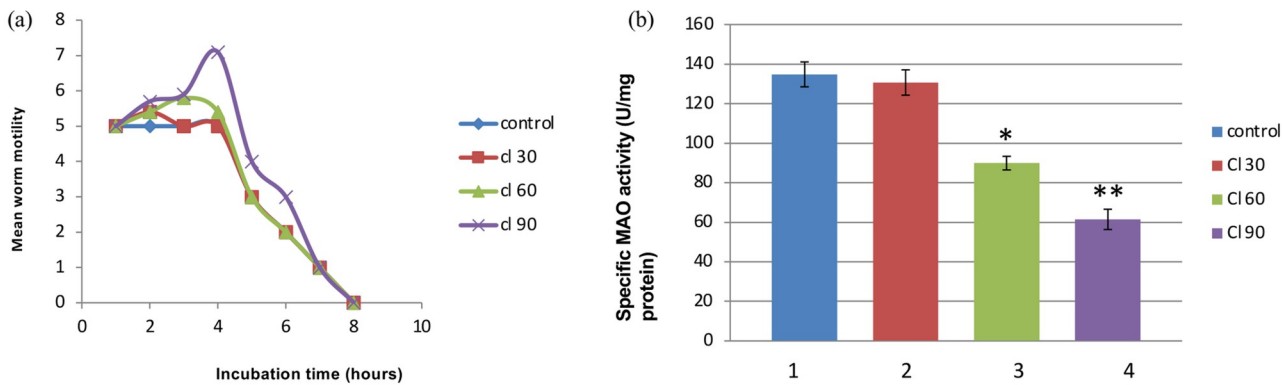

**Fig 6.** Motility of *F. gigantica* worms recorded at every 30 minutes over a period of 7 hours post incubation of parasites in different concentrations (30, 60 and 90μM) of clorgyline (a). Control was devoid of the specific inhibitor. In general, motility of worms upon administration of clorgyline first increased in a dose dependent manner and then decreased until complete immobility was evident. Grading system of motility was done by considering 1 as the least motile, 5 being highly motile and 7 being extremely motile. The specific enzyme activity of mitochondrial MAO-A in *F. gigantica* worms incubated *in vitro* with different doses of clorgyline (b). The adult *F. gigantica* worm treated with clorgyline at 30μM (2), 60μM (3) and 90μM (4) concentrations, while control (1) was devoid of any inhibitor. A minimum of 3 replicates were run and the data is expressed as a mean ± SEM. * $p < 0.05$, ** $p < 0.01$ were considered statistically significant.

MAO-A revealed that the native protein was catalytically highly active as evident from a high intensity band in the control samples. However, the enzyme activity decreased significantly following treatment with the parasite samples with clorgyline in a concentration dependant manner from 30 to 90μM. The 30μM concentration could not bring notable changes in MAO-A activity as compared to the control group but the highest dose of 90μM significantly

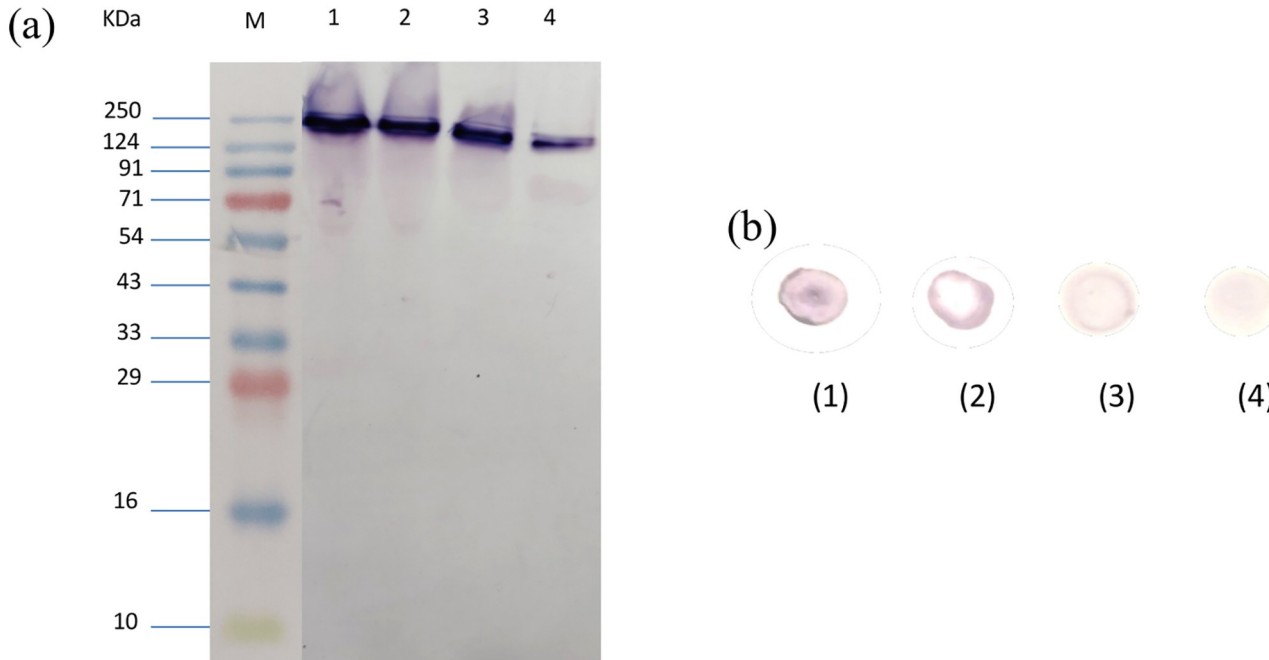

**Fig 7.** Representative image of the zymographic profile (a) of the mitochondrial MAO-A of *F. gigantica* on treatment with different concentrations of clorgyline (Lane 2: 30μM, Lane 3: 60μM and Lane 4: 90μM) and control without inhibitor (Lane 1). Dot blot assay of whole homogenate of *F. gigantica* samples treated with different concentrations of clorgyline (b) 1. Control 2. 30μM 3. 60μM 4. 90μM. All these results confirmed a concentration dependent decrease in the activity of MAO-A with an increase in the inhibitor dose from 30 to 90μM clorgyline.

**Table 1. The specific enzyme activity and percent inhibition of mitochondrial MAO-A enzyme in *F. gigantica* worms following treatment with different doses of clorgyline.**

| Clorgyline (μM) | Enzyme activity (μmol/min/mg protein) | Percent inhibition of enzyme activity (%) |
|---|---|---|
| 1. 0 (Control) | 134.78 | - |
| 2. 30 | 130.67 | 3.04 |
| 3. 60 | 89.90 | 33.29* |
| 4. 90 | 61.45 | 54.40** |

(1.) Control devoid of inhibitor (2.) The adult *F. gigantica* worms treated with clorgyline at 30μM. (3.) 60μM (4.) 90μM concentrations. A minimum of 3 replicates were run and the data is expressed as a mean ± SEM.

* $p < 0.05$,

** $p < 0.01$ were considered statistically significant.

inhibited the MAO-A activity as evident from the low intensity band, indicating that the enzyme activity was not completely lost even at the highest concentration of the inhibitor used in the present study (Fig 7a).

The Dot-Blot assay was performed considering the fact that it might provide future avenues for field-based detection of infection due to the simplicity of the test. The results revealed that the intensity of spots varied with the concentration of clorgyline. Spots with least dose of clorgyline i.e., 30μM showed highly intense spots and with higher doses of clorgyline i.e., 60μM and 90μM the spots appeared to be fading, thus revealing the inhibitory nature of clorgyline, and the results appeared to be consistent with zymography and also showed significant reproducibility (Fig 7b).

## Discussion

Helminthic infection in livestock is so detrimental that according to estimations by Spithill *et al.* [1] fasciolosis alone incurs economic losses of billions of USD annually. TCBZ is the most favored broad-spectrum drug used against helminthic infections for decades but the emergence of drug resistance has been observed which has prompted scientists across the world to find more alternatives of drugs as well as antigenic targets for their utilization in the development of successful vaccine. As non-conventional anthelmintics, our group has found varied efficacy in active components of natural products like curcumin and thymoquinone [18, 21], metalloenzyme inhibitors [59] and metal-based nanoparticles [60] in helminths of bovine origin. Though a plethora of work has been done in the direction of finding various antigenic targets, the exploration of neurobiological targets remained relatively untouched in helminths.

This is the first ever study on characterization of MAO using multipronged immunological approaches in *F. gigantica* thus filling the empty lacunae of understanding the relation between neurophysiology and its associated antigenic targets. Biogenic amines and their metabolizing enzymes have been reported in a number of parasitic helminths [49, 61–65] however conclusive studies regarding the activity, localization, biophysical properties of the enzyme in helminths are still in dearth.

First of all, identification of collected parasites was done on the basis of molecular characterization of ITS2 gene. The successful amplification of ITS2 gene using *F. gigantica* ITS2 specific primers suggested that the parasite under consideration was *F. gigantica* and not any other species of *Fasciola*. The ITS2 gene was selected for the identification of parasites, as these are highly conserved sequences and are considered as good marker for the identification purposes. The *F. hepatica* is more leafy-like in appearance with broad shoulders and smaller profile while *F. gigantica*, as the name suggests, is a bigger cousin of former and has narrower

shoulders and elongated body. Despite some morphological differences in *F. gigantica* and *F. hepatica* identification of parasite merely by visual means cannot be completely relied upon as there have been several reports of overlapping morphological characters in case of both the parasites and the existence of hybrid forms further complicate the proper selection of parasites for various studies. Therefore, to rule out the possibility of picking up a hybrid form from the field infection since previous reports have shown the occurrence of not only mixed infections of *Fasciola* species but also the occurrence of the hybrid forms, a thorough morphological identification was augmented by the molecular characterization of the tropical liver fluke species using *F. gigantica* specific ITS2 primers. The results revealed a successful amplification of the ITS2 gene using the end-point gradient-PCR method and the amplicons obtained thus further confirmed the identity of the specimens used in the biochemical and immunological studies.

Two types of MAO have been suggested in helminths where MAO-A is directly involved in the aggression and motility of the parasites and MAO-B is postulated to be responsible for general neuromuscular coordination [23]. So, to check the presence of two forms of enzyme in *F. gigantica*, specific enzyme activity was measured either alone or separately in the presence of Clorgyline and Deprenyl which are inhibitors of MAO-A and MAO-B respectively. A significant decline in the levels of enzyme activity when MAO-A and MAO-B inhibitors were used independently confirms the elementary presence of two isoforms of MAO in adult *F. gigantica* worms as also suggested in case of *G. explanatum* and *G. crumenifer* by Abidi and Nizami [23]. These results further support the previous studies on other digeneans such as *S. mansoni*, *S. cervi*, *A. galli*, *H. diminuta*, *A. lumbricoides* and *N. braziliensis* [38, 41, 42, 48, 66, 67] which showed the suppression of MAO activity by the action of varied number of inhibitors.

From this point our study was mainly focused on MAO-A of *F. gigantica*. as it is the more relevant isoform in relation to virulence and aggression of the parasite. Since MAOs are the enzymes located on the outer membrane of mitochondria, anticipation of increased enzyme activity in isolated mitochondrial samples found to be correct as the activity of enzyme was observed to be 1.5 times higher in the mitochondrial samples as compared to the whole homogenate of *F. gigantica*, as also described by Siddiqui and Podesta [68] and Schnaitman *et al.* [69] in *H. diminuta* and rat liver respectively. This signifies the high enrichment of mitochondria of *F. gigantica* worms with that of MAO-A enzyme.

A single prominent band of about 250KDa in the zymographic analysis indicates the large size of the enzyme in the native state whereas western blot analysis followed by the SDS-PAGE showed the single band size of about 50KDa which clearly signals towards the multimeric form of the enzyme of same sized monomers. These findings further strengthen the observations made by Dostert *et al.* [35]. When the zymography of whole homogenate was performed, a single band with low intensity was observed. The preliminary findings regarding the size of the enzyme in its active state and cleaved state could prove to be of immense value for further biochemical, pharmacological and *in silico* studies. The high enzyme activity as indicated by high intensity bands shows that the MAO-A is a major neurobiological enzyme which is required by the tissues in high amount for cellular housekeeping processes.

To determine the immunogenic potential of the enzyme, an immunodiagnostic approach was undertaken by means of Dot-Blot assay. The results of the Dot-blot assay using the whole homogenate of *F. gigantica* along with other trematodes revealed that the anti-MAO-A antibody used in the present study could detect the antigen with varied intensities of the dots in different species of the parasite. Amongst the bovine parasites, *F. gigantica*, *G. explanatum* and *G. crumenifer* and the ovine parasite, *P. epiclitum* were found positive but the *C. complanatum*, a parasite of piscine origin was found to be negative which suggests the presence of common and distinct epitopes of MAO-A enzyme. Such differences and similarities in the epitopes of

MAO-A could be in response to occupying different micro-environments in diversified hosts. The Enzyme Linked Immunosorbent Assay (ELISA) results further confirmed the strong antigenic nature of MAO-A in *F. gigantica* adult worms as evident from the high antibody titre at 1:6400 dilution. The minimum amount of the antigen which could be detected by the antibody used in the present study was found to be around 400ng, reflecting high level of sensitivity of detection process.

Since the MAOs are present in the outer membrane of mitochondria, immunolocalization studies revealed that MAO-A is located ubiquitously in the *F. gigantica* worms but its presence was more profound around the intestinal caecae, tegument and hypodermis. This is consistent with the previous studies in *A. galli* and *S. cervi* where MAO was found to be mainly concentrated in the CHM layer (cuticle, muscle, hypodermis) along with the intestine and gonads [38, 70]. Since the helminths mostly rely on parasitizing their host, their food also contains amines from the dietary sources of their host, the localization of MAO around the intestinal caecae seems to be justified as these MAOs nullifies the surplus biogenic amines which causes cellular toxicity [32]. The richness of tegument with MAO could be explained by the fact that a part of these parasitic worms comes from absorption of nutrients through body of host, so an outer barrier of checkpoint of the spurious monoamines is needed to avoid the toxicity.

A plethora of data is available regarding the effect of clorgyline, a MAO-A specific inhibitor on mammalian cells due to its effectiveness in the treatment of neurological diseases such as Alzheimer's and Parkinson's disease as well as in the treatment of clinical depression. However, with the exception of some preliminary data almost nothing is known about the effect of clorgyline on MAO-A of *F. gigantica* adult worms, in particular and helminths in general. Therefore, in the present study an attempt has been made to see the effects of clorgyline, a commercially available MAO-A inhibitor, on the activity and levels of the mitochondrial MAO-A of adult *F. gigantica* worms through a multitude of biochemical and immunological approaches. Parasites upon inhibitor administration showed a concentration dependant increase in the amplitude of motility during initial 3–4 hours in a 7-hour incubation period followed by the complete paralysis of the worms and eventually death of the worms. The initial increase in the amplitude of the motility of worms found to be directly proportional to the amount of clorgyline concentration as worms incubated with lower doses of clorgyline showed lower spike in motility in the initial phase of incubation followed by a continuous decrease in activity and eventually death. The results are in consensus with the previous studies on neuro-muscular coordinative role of MAO of parasites as described by Ercoli *et al*. [71] where the authors have shown that monoamine antagonist, methysergide, significantly affected the motility of *S. mansoni* worms in a dose dependant manner. In another study by Pax *et al*. [72] it was found that 5-HT, a prime MAO substrate, elevated the rhythmic contractibility in circular as well as longitudinal muscles. The incidental increase in motility of worms could be understood by the widely accepted nature of serotonin as a motility enhancer. As a result, when the level of MAO-A is suppressed by clorgyline, the possible upsurge of serotonin levels might have increased the motility of worms in the initial hours of incubation as also explained by Camicia *et al*. [73] who observed that citalopram, a serotonin transporter inhibitor, caused a significant inhibition in the motility of *E. granulosus*.

The activity of enzyme in clorgyline treated worms showed similar trend as that of the motility assay. The enzyme activity declined in a concentration dependant manner with an increase in inhibitor concentration. The significant inhibition of MAO-A in *F.gigantica* worms by such small doses of clorgyline (90µM) gives hopes in the development of drugs which can selectively target helminth associated MAO since the mammalian MAO is inhibited at much higher dose (10mg/kg body weight) as suggested by Green and Youdim [74]. This information could prove to be useful for the development of inhibitor-based drugs as lower

doses of inhibitory drugs against parasitic MAO, as compared to its mammalian host, may dislodge the parasite from its host rendering the host unharmed.

An overlapping trend with the previous results was obtained in zymography of clorgyline treated worms where the intensity of MAO-A bands was found to be continuously decreasing with the increase in clorgyline concentration which further confirmed the strong inhibitory nature of clorgyline against the MAO-A of parasitic origin. The enzymatic bands of clorgyline treated worms were exactly at the same position as it was in control group devoid of any inhibitor which gives supplementary affirmation that clorgyline do not cleave the MAO-A instead it binds to a site of enzyme which makes it difficult for substrate to bind with it thus reducing the overall activity of the enzyme. Though further studies are required to strengthen this claim.

The Dot-blot assay for *F. gigantica* samples treated with different concentration of clorgyline using anti-MAO-A primary antibody revealed that the intensity of spots decreased in a similar manner as the band intensity in zymography. These findings are in agreement with earlier studies of Agarwal *et al.* [70] who observed that clorgyline caused significant inhibition of MAO levels in the neurotropic filarid, *S. cervi*. A marked decline in the immunogenic potential of MAO-A by clorgyline signalled towards a strong binding affinity of this inhibitor with the immunogenic epitopes of the enzyme. Also, the immunodetection of MAO-A by Dot-blot assay in the whole homogenate of *F. gigantica* samples makes it a good candidate for the development of a simple diagnostic tool to detect the field-based infection, of *F. gigantica*.

## Conclusions

Considering the present results, it can be elucidated that MAO-A is an important neurophysiological enzyme having a high molecular mass of about 250KDa and is abundantly present in the mitochondria of tropical liver fluke, *F. gigantica*. The enzyme is ubiquitously present in the fluke but is highly concentrated around the intestinal caecae and tegument felicitating cytotoxicity prevention by surplus monoamines absorbed through these nutrient absorptive surfaces. Also, the enzyme MAO-A found to be highly immunogenic biomolecule. Owing to its importance in the survival and perpetuation of helminths, it could be a relevant target for future drug designing as well as in the development of vaccines. However, further studies are required to understand the mechanism and role of MAO-A in virulence as well as in the establishment of host-parasite relationship. Clorgyline caused a significant inhibition of MAO-A activity in adult *F. gigantica*, initially increasing the instantaneous motility of the worms, which ultimately led to the complete paralysis of worms as that might lead to the clearance of parasites from the body of the host. Therefore, further studies should be undertaken to explore the therapeutic effectiveness of clorgyline in order to develop an alternative drug formulations / flukicides in order to control the liver fluke infections.

## Supporting information

**S1 Fig. Zymography of MAO-A in *F. gigantica* whole homogenate isolate (lane 1 and 2) showing MAO-A.** Note: Lane 1 is not the part of main figure and Marker in all zymography gels appeared to be very faint.
(TIF)

**S2 Fig. Zymography of MAO-A of *F. gigantica* on treatment with different concentrations of clorgyline (lane 3: 30μM, lane 4: 60μM and lane 5: 90μM) and control without inhibitor (lane 1 and lane 2).** Note: Marker in all zymography gels appeared to be very faint.
(TIF)

**S3 Fig. Western blot of MAO-A in *F. gigantica* adult worms (lane 1 and 2).** Note: Lane 2 is not the part of main figure.
(TIF)

## Acknowledgments

The authors are grateful to Chairman, Department of Zoology. Aligarh Muslim University, Aligarh for providing necessary facilities and Coordinator, University Sophisticated Instrumentation Facility for confocal microscopy. We are extremely thankful to CSIR, UGC, DBT-BUILDER and BBSRC for instrumentation support. Technical assistance of Mr. Sarfaraz Ahmad is also greatly acknowledged.

## Author Contributions

**Conceptualization:** Mirza Ahmar Beg, S. M. A. Abidi.

**Data curation:** Mirza Ahmar Beg, Abdur Rehman.

**Formal analysis:** Mirza Ahmar Beg, Abdur Rehman, Lubna Rehman, Rizwan Ullah.

**Investigation:** S. M. A. Abidi.

**Methodology:** Mirza Ahmar Beg, Lubna Rehman, Faiza Farhat, Sobia Wasim.

**Resources:** S. M. A. Abidi.

**Supervision:** S. M. A. Abidi.

**Validation:** S. M. A. Abidi.

**Writing – original draft:** Mirza Ahmar Beg.

**Writing – review & editing:** Mirza Ahmar Beg, Abdur Rehman, S. M. A. Abidi.

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
