## [Decision Letter · Decision Letter 0]

8 Mar 2023

PONE-D-22-30479Characterization of Monoamine Oxidase-A in tropical liver fluke, Fasciola giganticaPLOS ONE

Dear Dr. Beg,

Thank you for submitting your manuscript to PLOS ONE. After careful consideration, we feel that it has merit but does not fully meet PLOS ONE’s publication criteria as it currently stands. Therefore, we invite you to submit a revised version of the manuscript that addresses the points raised during the review process.

We look forward to receiving your revised manuscript.

Kind regards,

Paulo Lee Ho, Ph.D.

Academic Editor

PLOS ONE

“No funding was available for this project”

Reviewers' comments:

Reviewer's Responses to Questions

**Comments to the Author**

1. Is the manuscript technically sound, and do the data support the conclusions?

Reviewer #1: No

Reviewer #2: Yes

2. Has the statistical analysis been performed appropriately and rigorously? 

Reviewer #1: No

Reviewer #2: Yes

3. Have the authors made all data underlying the findings in their manuscript fully available?

Reviewer #1: Yes

Reviewer #2: Yes

4. Is the manuscript presented in an intelligible fashion and written in standard English?

Reviewer #1: Yes

Reviewer #2: Yes

5. Review Comments to the Author

Reviewer #1: The manuscript is interesting, and important to the field since Fasciola sp infection is difficult to do a diagnosis because of the low presence of eggs of the parasite in feces and treatment is also problematic. So, it is important to discover new tools for diagnosis and treatment. Despite that, the manuscript has some problems that should be improved. The questions are:

Major:

1. In figure 1a, Where are the positive and negative controls of PCR? What is the size of the amplicon? Is it specific? Was it sequenced?

2. In figure 2b, did the researchers apply any statistical approach in this experiment?

3. In figures 3a and 3b how can I prove that is a mitochondrial protein extract? I suggest a control using an antibody against mitochondrial marker protein.

4. In this phrase "For the immunological characterization of MAO-A in F. gigantica worms a number of studies were performed". How many? Please specify, the number of studies made.

5. In figure 4b, where is lane 2?

6. The results which are shown in figure 7 sound interesting, but the authors should provide loading controls ( ponceau stained membrane and/or a mitochondrion antibody marker with present no alteration with inhibitor treatment. Without this, is not possible to do the affirmation made by the authors in the text.

7. It seems that the anti-MAO-A primary antibody was purchased from Sigma Aldrich. This antibody was raised in which organism? Fasciola? If not raised in Fasciola, what is the degree of identity of the epitope?

Minor:

1. In the phrase in the Introduction section: "Till now, MAO has been reported in parasitic helminths belonging to different classes, namely", please change "till" to "Until".

2. In Materials and methods in the PCR section: Please provide PCR reaction components concentrations

3. Please provide a link to the PCR amplification with other results.

Reviewer #2: PONE-D-22-30479

Title: “Characterization of Monoamine Oxidase-A in tropical liver fluke, Fasciola gigantica.”

Minor Revision

The publication does not overstate anything and addresses the titular claims well. It is a well referenced manuscript and authors have nicely mentioned drug resistance story and explained the present gaps in the characterization of MAO in helminths. Furthermore, characterization through multi-omics approach validates most of the findings. I believe molecular characterisation of parasite is a strong pivot of this manuscript because identification of field-procured parasite just on the basis of morphological identification could give false positive results. Although, I have no doubt on experimentation but the quality of manuscript can be improvised by small changes which I have enumerated below:

1. It is mentioned that MAO activity in mitochondria is significantly higher compared to whole homogenate but there is no significant asterisks on activity bars. Kindly put significant marks in Fig 2a.

2. Authors have forgotten to put lane 2 western blot in Fig 4b. Kindly fix that.

3. In Abstract, abbreviate the full form of ‘WHO’ as ‘World Health Organization’ because it is not a good idea to introduce abbreviated words in abstract because it is the first text the audience reads.

4. Almost at all the places spelling of ‘Dependent’ is wrong. Kindly replace “Dependent” with “Dependant”. For spellings, stay with either UK English or US English in a manuscript and not a mixture of both in a single publication.

5. Avoid using same adjective in a single sentence such as in this statement “The strong intensity of bands in western blots and the spots in Dot-blots indicate strong immunogenic nature of the MAO protein” strong is used two times as an adjective. Then again after two three lines “clearly indicating that the tropical liver fluke possesses strong MAO-A activity” the use of word “strong” comes into play. Kindly replace it with different synonyms.

6. The statement in 3rd paragraph of introduction “thus making MAO an indispensable biomolecule for developing the host-parasite interaction for the survival and perpetuation” does not make any sense or it is too complex. Kindly use simpler text for a better understanding for a wider range of audience.

7. Kindly rephrase the last paragraph statement in introduction “Previously researchers have reported and localized MAO in helminth parasites but data is still in dearth about the complete picture of this enzyme in a specific parasite”.

8. In the conclusive remarks “Considering the present results, it can be elucidated that MAO-A is an important neurophysiological enzyme having a high molecular mass and is abundantly present in the mitochondria of tropical liver fluke, F. gigantica” refrain from using unspecific words such as “high molecular mass” instead of that mention the size of band observed.

6. PLOS authors have the option to publish the peer review history of their article (what does this mean?). If published, this will include your full peer review and any attached files.

Reviewer #1: No

Reviewer #2: **Yes: **Dr. Shahanwaz Rehman

---

## [Author Response · Author response to Decision Letter 0]

7 Apr 2023

First and foremost, we sincerely thank the reviewers for putting up their valuable time in reviewing this manuscript. We thoroughly went through the comments made by reviewers and have corrected our manuscript as advised. The corrections and explanations behind those changes are enumerated as follows:

Reviewer#1

Query: Reviewer requested to include PCR reaction volume and concentrations in text.

Response: PCR reaction volume and concentrations of each of its ingredient are now provided in materials and methods section (Page 6 under heading “Amplification of DNA using Polymerase Chain Reaction”).

Query: Reviewer questioned about the statistical test done, if any, in figure 2b.

Response: The non-parametric t-test was performed for figure 2a and One-way ANOVA was performed for figure 2b. The level of significance along with their significance asterisk have now been included in the graphs as advised (Figure 2).

Query: Reviewer had advised to provide details about host of antibody and specificity of epitopes of antibody. 

Response: It is not possible to raise antibody in Fasciola gigantica because it does not have well defined circulatory system. The antibody was purchased from Sigma-Aldrich which was raised in mouse with the epitopes of human MAO and cited reactivity with mouse samples. Since the MAO is highly conserved protein and with its reactivity with mouse we hypothesized it should react with Fasciola samples too. No antibody of MAO against any parasite was available in the market.

Query: Reviewer had requested to mention specificity of primers and size of amplicons obtained in ITS2 based molecular identification experiment. He had also advised to include positive and negative control in PCR. 

Response: As suggested, we have included negative control of PCR in our results. Since the experiment was done for the first time it was not possible to have positive control. Regarding the size of amplicon, it is in consensus with previously reported Fasciola gigantica ITS2 amplicons (300-500 bp) which were used by other researchers for molecular characterization of F. gigantica in their area of study. The primers used were specific for Fasciola gigantica the details of which is included in materials and methods section (Page 6 under the heading Amplification of DNA using Polymerase Chain Reaction).

Query: Reviewer sought an explanation on successful extraction of mitochondria.

Response: Here, we were not proving that MAO-A is localized in mitochondria. It is already an established fact that MAO-A is present in mitochondria and the substrate we used in our study for detection was specific for MAO-A enzyme which clearly explains why we saw significantly higher signals in mitochondrial extracts compared to whole homogenate.

Significantly high activity of MAO-A in mitochondrial extract compared to whole homogenate samples in itself shows that there was successful extraction of mitochondria which we did following well-proven method of Podesta et al. [50].

Query: Reviewer had asked authors to include number of studies made in response to the statement “For the immunological characterization of MAO-A in F. gigantica worms a number of studies were performed”.

Response: By this statement "For the immunological characterization of MAO-A in F. gigantica worms a number of studies were performed" we were not referring to the number of replicates. Our intent was to say that we adopted multipronged and multi-omics approach for the characterization of MAO such as biochemical, immunological and pharmacological for a profound validation of results. 

Query: Reviewer pointed out that there is no lane 2 in figure 4b.

Response: We realised that legend of figure 4b was wrongly written and the legend is now corrected as suggested (Page 14, caption of Fig 4).

Query: Reviewer had asked about loading control in clorgyline treated western blotting result.

Response: Reviewer has raised a valid point of loading control but since the authors involved in this study have mov¬¬ed out of that lab it is not possible to replicate this experiment. Therefore, we have removed the western blot of clorgyline treated samples (Figure 7).

Query: Reviewer requested to change “Till now” to “Until now”.

Response: In the phrase " Till now, MAO has been reported in parasitic helminths belonging to different classes, namely ", till is changed to until (Page 5, second-last paragraph).

Reviewer#2

Query: Reviewer had requested to put level of significance in Figure 2. 

Response: Asterisks for level of significance following statistical tests has now been added to the graphs (Fig 2).

Query: Reviewer pointed out that legend and figure 3b is not in accordance with each other.

Response: The legend and figure are now corrected and lane2 is removed from legend (Fig 3).

Query: Reviewer had requested to write full form of WHO in abstract.

Response: WHO has now been abbreviated in full as World Health Organisation in Abstract (Page 1).

Query: Reviewer had asked to correct spelling of Dependant at all places in text.

Response: The spelling of Dependent has now been changed to Dependant at all places (Page 2, 4, 15, 16 and 21).

Query: Reviewer had pointed out some grammatical errors in Abstract and advised to correct it.

Response: The repetition of adjectives has been replaced as advised (Page 2, last para).

Query: Reviewer had requested to make changes to statement “thus making MAO an indispensable biomolecule for developing the host-parasite interaction for the survival and perpetuation”

Response: The text “thus making MAO an indispensable biomolecule for developing the host-parasite interaction for the survival and perpetuation” has been reframed to make more sense (Page 4, para 1). 

Query: Reviewer had asked authors to correct the statement “Previously researchers have reported and localized MAO in helminth parasites but data is still in dearth about the complete picture of this enzyme in a specific parasite”.

Response: The sentence “Previously researchers have reported and localized MAO in helminth parasites but data is still in dearth about the complete picture of this enzyme in a specific parasite” has been reframed as suggested (page 5, last para).

Query: Reviewer had requested to include size of band in conclusion instead of high molecular mass.

Response: In conclusion section, we have mentioned size of band as requested (Page 23).

The authors would like to sincerely thank the reviewers for their valuable inputs in making

our manuscript better. We hope that the revised version of the manuscript could now be

accepted for publication in Plos One journal.

---

## [Decision Letter · Decision Letter 1]

13 Apr 2023

Characterization of Monoamine Oxidase-A in tropical liver fluke, Fasciola gigantica

PONE-D-22-30479R1

Dear Dr. Beg,

We’re pleased to inform you that your manuscript has been judged scientifically suitable for publication and will be formally accepted for publication once it meets all outstanding technical requirements.

Kind regards,

Paulo Lee Ho, Ph.D.

Academic Editor

PLOS ONE

Additional Editor Comments (optional):

Reviewers' comments:

Reviewer's Responses to Questions

**Comments to the Author**

1. If the authors have adequately addressed your comments raised in a previous round of review and you feel that this manuscript is now acceptable for publication, you may indicate that here to bypass the “Comments to the Author” section, enter your conflict of interest statement in the “Confidential to Editor” section, and submit your "Accept" recommendation.

Reviewer #2: All comments have been addressed

2. Is the manuscript technically sound, and do the data support the conclusions?

Reviewer #2: Yes

3. Has the statistical analysis been performed appropriately and rigorously? 

Reviewer #2: Yes

4. Have the authors made all data underlying the findings in their manuscript fully available?

Reviewer #2: Yes

5. Is the manuscript presented in an intelligible fashion and written in standard English?

Reviewer #2: Yes

6. Review Comments to the Author

Reviewer #2: The manuscript is very well written, depicting the complete story in a lucid manner. The queries and comments are very well answered and introduced in the revised manuscript.

Thank You

7. PLOS authors have the option to publish the peer review history of their article (what does this mean?). If published, this will include your full peer review and any attached files.

Reviewer #2: **Yes: **Dr. Shahnawaz Rehman

---

## [Editor Report · Acceptance letter]

18 Apr 2023

PONE-D-22-30479R1 

Characterization of Monoamine Oxidase-A in tropical liver fluke, *Fasciola gigantica*

Dear Dr. Beg:

I'm pleased to inform you that your manuscript has been deemed suitable for publication in PLOS ONE. Congratulations! Your manuscript is now with our production department. 

Kind regards, 

on behalf of

Dr. Paulo Lee Ho 

Academic Editor

PLOS ONE